# Dingo Optimization Based Cluster Based Routing in Internet of Things

**DOI:** 10.3390/s22208064

**Published:** 2022-10-21

**Authors:** Kalavagunta Aravind, Praveen Kumar Reddy Maddikunta

**Affiliations:** School of Information Technology and Engineering, Vellore Institute of Technology and Engineering, Vellore 632014, India

**Keywords:** IoT, healthcare, fault tolerance, energy hole, SADO-BM scheme

## Abstract

The Wireless Sensor Network (WSN) is a collection of distinct, geographically distributed, Internet-connected sensors, which is capable of processing, analyzing, storing, and exchanging collected information. However, the Internet of Things (IoT) devices in the network are equipped with limited resources and minimal computing capability, resulting in energy conservation problems. Although clustering is an efficient method for energy saving in network nodes, the existing clustering algorithms are not effective due to the short lifespan of a network, an unbalanced load among the network nodes, and increased end-to-end delays. Hence, this paper proposes a novel cluster-based approach for IoT using a Self-Adaptive Dingo Optimizer with Brownian Motion (SDO-BM) technique to choose the optimal cluster head (CH) considering the various constraints such as energy, distance, delay, overhead, trust, Quality of Service (QoS), and security (high risk, low risk, and medium risk). If the chosen optimal CH is defective, then fault tolerance and energy hole mitigation techniques are used to stabilize the network. Eventually, analysis is done to ensure the progression of the SADO-BM model. The proposed model provides optimal results compared to existing models.

## 1. Introduction

The Internet of things (IoT) has become an necessary principle of human life. It is one of the modern era’s growing technologies that has drawn interest from both academics and industry [1]. All IoT devices communicate information via the Internet. IoT refers to a collection of various Internet-connected gadgets that may communicate with one another without the need for human involvement. Several IoT applications are available, such as smart homes, smart cities, smart agriculture, and smart healthcare. In the Wireless Sensor Network (WSN), numerous independent, geographically dispersed devices can wirelessly detect changes and send the information to the nearby base-station [2]. A sensor node is a device that can detect, process, and send sensor data from one node to another node. Based on the application requirements, such as the quantity of sensor nodes, node deployment, and power consumption, the WSN faces design issues. In networks with limited resources, energy conservation is a top priority; as a result, optimal routing plays a major role in providing energy optimization. Routing can be carried out in several ways like choosing optimal network organization, finding optimal route discovery, and protocol operation in accordance with the deployment of the network’s nodes [3].

The combination of IoT along with cloud-oriented applications provides better results compared with traditional cloud applications in terms of efficiency [4]. The cloud-based IoT offers various medical services like continuous live monitoring, providing notifications, and accessing health records at minimal time [5,6]. The embedded senors capture the patient’s health parameters, identify the patient’s health condition, and send the notifications to the health care providers [7,8]. However, the IoT devices which are employed in smart health applications have limited processing and battery storage. As the IoT sensors capture the data continuously, this leads to faster draining of the device’s battery, which results in poor network performance [9]. Thus, there is a need to improve the network performance and increase the life time of the network energy optimization in IoT. Many researchers have proposed several techniques for optimizing the battery life of IoT devices, and clustering is considered as one of the efficient techniques in optimizing the energy. Table 1 lists the abbreviations used in this study.

Clustering helps in resolving some of the typical problems in the IoT environment such as increasing the energy efficiency, improving the scalability, minimizing the dependability, and increasing the network lifetime. The group of sensor nodes in a network are formed into a cluster, and the optimal node in a cluster is considered as cluster head (CH). The CH is chosen based on various parameters like residual energy, distance, delay, overhead, QoS, and trust. All the live nodes in a cluster transfer the information to their respective CH. The CH is responsible for transferring the aggregated data to the sink node. During this process, the CH loses its energy. In this regard, the selection of optimal CH has to be done in an iterative manner. In WSN, the Low-Energy Adaptive Clustering Hierarchy (LEACH) protocol is a common clustering protocol. However, LEACH suffers unequal distribution of clusters, owing to the random placements of nodes in the network; therefore, cluster members (CMs) are unable to send the data to the sink due to the failure of CH node by the hotspot problem. The Hybrid Energy-Efficient Distributed Clustering (HEED) protocol suffers from an overhead problem, which leads to data redundancy. Some of the latest algorithms face several challenges such as:1.High end-to-end delay.2.More consumption of energy.3.Minimum residual energy.4.Minimum network life time.5.Fewer alive nodes.6.Poor node failure management.

In order to overcome all these challenges, the present work focuses on the following contributions:Deploying SADO-BM for optimally electing the CH in the IoT network.Optimal alive nodes; increasing the residual energy; optimal convergent rate; providing optimal delay and distance; increasing the network life time.Exploiting fault tolerance and the energy hole mitigation process if the optimal CH is found to be a defective one.

The remainder of this paper is organized as follows. Section 2 presents a summary of related works. Section 3 describes the general idea on IoT in healthcare appliances. Section 4 and Section 5 present the objectives and development of SADO-BM. Section 6 describes the fault tolerance and energy hole mitigation. The results and conclusions are given in Section 7 and Section 8.

## 2. Literature Review

### 2.1. Related Works

In 2021, Banyal et al. [10] developed a novel technique to segment the network topology as per the node’s features. This model was achieved via “intelligent transmission”. The recommended scheme of the hierarchical learning-based sectionalized routing (HiLSeR) was employed for routing the packets. For “topology sectionalization and routing decision making, hierarchical learning, a multi-dimensional data conduct oriented soft clustering paradigm”, was employed. While carrying out tests, the effectiveness of the projected model was assessed over other schemes. To demonstrate the improved effectiveness, varied constraints such as “Energy Unit per Message, Dead node Percentage, Overhead Ratio, Average Latency, and Success Ratio” were calculated.

In 2019, Shreshth et al. [11] suggested a new scheme called Health-Fog that incorporated “ensemble deep learning in Edge computing” and employed it as a practical appliance of automated disease study. Health-Fog conveyed medicinal care via IoT and efficiently handled patients’ data that arrived as user requests. Additionally, FogBus was employed to test the performance of the suggested scheme regarding power utilization, accuracy, implementation time, latency, and bandwidth and jitter.

In 2020, Bharathi et. al. [12] offered an Energy Efficient Particle Swarm Optimization (EEPSO) technique geared toward capable CH assortment amid wide-ranging IoT devices. The IoT device employed to find medicinal data was clustered and a CH was elected using EEPSOC. The elected CH forwarded the data to cloud servers. Consequently, the CH was liable to convey data of IoT to the cloud using fog devices. Subsequently, an Artificial Neural Network (ANN) classifier was deployed to diagnose the medicinal data in the cloud to recognize the severity of the disease.

In 2020, Naghibi et al. [13] modeled a method to divide the network into definite cells in a geographical manner and used two mobile sinks to collect the information noticed by cell nodes. As per the communication amid cells and mobile sinks, the cells were detached into two modules: “Single-Hop Communication Cells (SCCs) and Multi-Hop Communication Cells (MCCs)”. When the sinks are stationary, SCCs broadcast information to the sinks straight away; nevertheless, MCCs adopted the Energy Efficient Geographic Routing Protocol based on Mobile Sink (EGRPM) scheme to broadcast information to sinks.

In 2018, Hao et al. [14] presented a novel Energy-Efficient Localization (EEL)–oriented geographic routing model that used locality data and the remaining energy of SNs for forwarding the data packets to sink nodes. At regular intervals, EEL updated the locality info of Sensor Nodes (SNs) in an Underwater Wireless Sensor Network (UWSN) and efficiently adapted to the vibrant network topological variations. EEL iterated via a set of forwarding nodes of candidates by considering Normalized Advancement (NADV), which determined the priority level of transmission. The simulated results showed that EEL efficiently located SNs while considerably improving the Packet Delivery Ratio (PDR) and reducing the energy utilization in the routing procedure.

In 2020, Hadikhani et al. [15] developed a distributed scheme which rapidly determined and updated the boundary of a hole. Per hole, the packets were directed in a flee path about the hole that improved and extended the network life span. The results of tentative simulations were evaluated over successful and renowned routing techniques in WSN. This evaluation has shown that the anticipated scheme reduced dead nodes. In addition, the network life span was enhanced regarding time and node count.

In 2020, Ghaderi et al. [16] proposed a protocol dependent on the fuzzy technique. Here, the sensor areas were divided into effective hexagonal grid cells, and after that, the cells were laid based on geographic location. Then, in every sampling round, CH sensors in every grid cell were elected depending upon the fuzzy model. After that, the CH reading was conveyed to sink in a multi-hop path per the fuzzy model. Simulated results demonstrate that fuzzy offered better effectiveness over other techniques.

In 2019, Vahabi et al. [17] proposed an amalgamation of hierarchical and geographic techniques with mobile sinks to decrease energy utilization and increase the network life span. With this technique, the remaining energy was amplified, thus significantly increasing the network life span. Results of the investigations showed that the suggested scheme increased the network life span over other schemes.

### 2.2. Review

Table 2 reviews the extant IoT protocols. Primarily, [10] used the HilSeR scheme to contain high network energy, utmost alive nodes, lesser latency, and lesser traffic volume; nevertheless, the PDR was low. The bagging classifier used in [11] offered higher accurateness and negligible implementation time. On the other hand, cost-optimal implementation was not measured. ANN used in [12] offered improved specificity with improved accuracy; however, compressive sensing was not studied. The authors of [13] used the EGRPM to contain a superior life span and lesser delay; however, there was a higher overhead. Energy-Efficient Localization (EEL) routing used in [14] provided higher PDR and lesser error; however, there was more energy consumption. In addition, Efficient Load Balanced Routing (ELBAR) used in [15] presented a better lifespan with less energy utilization, but it failed to spotlight numerous holes. Fuzzy Geographic Routing Protocol Based On Compressive Data Gathering (FGAF CDG) used in [16] incurred negligible distance amid hops and low energy utilization; nonetheless, it required evaluation on communication costs. Lastly, IoGHR used in [17] offered higher life span with high residual energy; however, several mobile sinks should be deployed in future study.

## 3. Idea on IoT in Healthcare Appliance towards Cluster-Based Routing

This work focuses on the clustering model in IoT for medicinal appliances. Usually, there are two types of Electronic Health Record (EHR) users: “(a) patients or EHR owners, and (b) EHR users who are not owners, but can be a health insurance company, physicians, researcher, family members or friends of patients, pharmacists or doctors”. Patients (EHR owners) are endorsed to upload the encrypted EHR to IoT by allowing the user’s access to specified EHR parts. The data are then stored in IoT servers. The EHR owners depict the accessing levels to each user in the ACL. The owner may also proffer total access to EHR, primarily to the patients’ closer associates or relations. Figure 1 shows the representation of the proposed model.

Step 1: First, the EHR owners are registered in the IoT server.

Step 2: After registering, they upload the records into the server.

Step 3: If a customer (for example, a friend of the patient, a doctor, or pharmacist) requires accessing the EHR of patients, the EHR administrator confirms their access level to transfer the record.

This work deploys SADO-BM for selecting CH by considering “energy, distance, delay, overhead, trust, QoS, and security (high risk, low risk, and medium risk)”. If the selected CH has any defects, fault tolerance and energy hole mitigation are then performed.

Those are the steps followed during data transmission; however, which data to be transferred through which route is also a major question as it needs quick response as much as possible due to emergency cases. For proper routing, a strategy is needed across the network. Therefore, most of the routing algorithms follow routing with different constraints. In this way, the proposed model ensures secure data routing that includes the constraints like Energy, Delay, Distance, Trust Model, QoS, and Security.

By considering this as the optimization issue, the problem is solved by a new SADO-BM algorithm. Moreover, fault tolerance and energy hole mitigation are carried out in case of defects in CH.

## 4. Objectives and Description

The intention of the SADO-BM based scheme for electing the optimal CH is presented in Equation (Equation 1).
(1)$$Obj=\min\left[\begin{array}{c} \left(we_{1}*En\right)+\left(we_{2}*Dis\right)+\left(we_{3}*Dl\right)+\left(we_{4}*OH\right)\\ +\left(we_{5}*\left(1-Tr\right)\right)+\left(we_{6}\left(1-QOS\right)+we_{7}\left(1-se\right)\right) \end{array}\right]$$

In Equation (Equation 13), $we_{1}$—$we_{7}$ are weight factors calculated using a tent map.

### 4.1. Energy

It is very important to decide the life span of the network. The battery cannot be re-energized because there is no power source, and transferring data to BS requires more energy. In Equation (Equation 2), $En\left(P_{l}\right)$ implies the energy of *l*th hop, and *di* indicates the hop count for multi-hop routing. “The energy consumed during communication $En\left(P_{l}\right)$ is in the form of energy required for transmitting packets $En_{TX}$, receiving the packet $En_{RX}$, at idle state $En_{1}$ and energy cost $En_{ST}$”.
(2)$$\;\mathrm{Energy}\;=\frac{1}{di}\sum_{l=1}^{di}En\left(\rho_{l}\right)$$
(3)$$En=En_{TX}+En_{RX}+En_{1}+En_{ST}$$

The energy used for transmitting packets ($En_{TX}$), electronic energy ($En_{ete}$), energy for data compilation ($En_{agg}$), and threshold energy ($en_{0}$) are represented in Equations (4)–(7), respectively; here, “$E_{TX}\left(M:en\right)$ implies the energy essential to communicate *m* bytes over *en*th distance”.
(4)$$En_{TX}\left(M:en\right)=\left\{\begin{array}{c} En_{ete}*M+En_{fr}*M*en^{2},\;\mathrm{if}\;en<en_{0}\\ En_{ete}*M+En_{pr}*M*en^{2},\;\mathrm{if}\;en\;\geq en_{0} \end{array}\right.$$

Here,
(5)$$En_{ete}=En_{TX}+En_{agg}$$
(6)$$En_{agg}=En_{fr}en^{2}$$
(7)$$en_{0}=\sqrt{\frac{En_{fr}}{En_{pr}}}$$
In Equation (Equation 7), $En_{pr}$ implies “power amplifier energy and $En_{fr}$ implies energy vital to deploy free-space system”.

### 4.2. Delay

Delay is a noteworthy QoS factor to forwarding data. “It is known as the hope ratio necessary for the total number of routing nodes in the network” and is presented in Equation (Equation 8), in which *d* implies the distance traveled.
(8)$$Dl=\frac{d}{speed}$$

### 4.3. Distance

The distance (dis) among nodes is an imperative factor that portrays the network’s life span. It is represented in Equation (Equation 9), in which *v* implies the speed of SN, and *t* implies time.
(9)Dis=v×t

### 4.4. Trust Model

Every network hop comprises a superior trust degree to assess the trust amid the relevant nodes and hops near to it. There are are kinds of trust: “(i) Direct trust; (ii) Indirect trust” as presented in Equation (Equation 10).
(10)$$Tru=\left\{Tru^{D}+Tru^{I}\right\}$$

(i) Direct Trust ($Tru^{D}$): “The direct trust is known as local trust and it presents the trust value as an agent to determine the familiarities with the target agent”. It is modeled in Equation (Equation 11), wherein, $B_{v1,v2}\left(t\right)$ is the appropriately broadcasted packet count via SN $v_{2}$ to $v_{1}$ at *t*. Moreover, $C_{v1,v2}\left(t\right)$ is the packet count broadcasted by SN $v_{2}$ to $v_{1}$ at *t*.
(11)$$Tru^{D}\left(t\right)=\frac{B_{v1,v2}\left(t\right)}{C_{v1,v2}\left(t\right)}$$

(ii) Indirect Trust ($Tru^{l}$): “It is determined from the knowledge obtained through other hops. The knowledge of other hops helps in decide each transaction”. It is modeled in Equation (Equation 12), in which *q* is the adjacent node count.
(12)$$Tru^{I}\left(t\right)=\frac{1}{q}\sum_{n=1}^{q}Tru^{D}\left(t\right)$$

### 4.5. QoS

The *QoS* is the method for controlling the network sources to diminish packet loss, latency, and network jitters. The *QoS* is precisely modeled in Equation (Equation 13), in which, *R* pertains to node safety.
(13)$$QoS=mean\left(R\right)$$

### 4.6. Security

This feature includes three modes that are described as follows:

Security mode: The security mode operation selects the CH which meets the requirement of security. In Equation (Equation 14), $s_{r}$ and $s_{s}$ refer to the security requirements related to Cluster Head Selection (CHS) and security rank. If $s_{s}$≤$s_{r}$, it can be said to be the required CH. Additionally, choosing the cluster head among diverse nodes requires a cautious approach that can be termed as a secure mode.

Risky mode: In this mode, an extant CH is chosen to obtain optimal CH in order to capture all risks. Thus, this mode is termed as the “insistent mode during the CHS procedure”.

γ-risky mode: The CH with high risks is preferred in γ-risky mode. Moreover, γ-risk is termed as $U_{risk}$ and it includes two values, namely, γ = 0 and γ = 1. In addition, “if the chosen CH achieves the state $s_{s}$ > $s_{r}$ the risk should be less than 50%. If the condition is $0<\zeta_{s}-g_{r}\leq 1$, the selection process would be implemented, and if the state is $1<\zeta_{s}-g_{r}\leq 2$, there would be a delay in the selection process. However, the CHS process would not be completed, and the corresponding function should be continued for the state $2<\zeta_{s}-g_{r}\leq 5$”.
(14)$$se=u_{risk}=\left\{\begin{array}{ccc} 0 & \;\mathrm{if}\; & \zeta_{s}-\zeta_{r}\leq 0\\ 1-e^{\frac{\left(s_{x}-\zeta_{r}\right)}{2}} & \;\mathrm{if}\; & 0<\zeta_{s}-\zeta_{r}\leq 1\\ 1-e^{\frac{3\left(\zeta_{s}-\zeta_{r}\right)}{2}} & \;\mathrm{if}\; & 1<\zeta_{s}-\zeta_{r}\leq 2\\ 1 & \;\mathrm{if}\; & 2<\zeta_{s}-\zeta_{r}\leq 5 \end{array}\right.$$

### 4.7. Overhead

In IoT, the broadcast of packets lead to overhead and as a result, it is necessary for communication. Message monitoring and header length should be lessened, since they cause connectivity cost. The growing number of routing packets swapping all through the simulation is called overhead and it is implied as *OH*.

## 5. Developed SADO-BM for Optimal CHS

Solution encoding: As mentioned earlier, CHs are optimally chosen via the SADO-BM model. The representation for solutions is shown in Figure 2, which reveals the count of CHs.

### Proposed SADO-BM Model

DOX [18] is a well-known optimization model with enhanced convergence. Nevertheless, to advance the searching excellence, some improvisation is essential. Self-improvisation is superior for optimization issues. The numerical portrayal of SADO-BM is described here.

Step 1: Initialize the *N* populace *pop* of searching agents. Moreover, initialize the utmost iteration $max^{itr}$. The present iteration *itr* is fixed as 0.Step 2: The initial searching agents are $D_{iN}$.Step 3: The value of $\bar{b}$,$\bar{A}$,$\bar{B}$ are initialized. The encircling act of DOX is presented in Equation (Equation 15), in which $D_{d}$ refers to the distance between the dingo and prey; $P_{p}$ is the Positioning of a prey vector; *P* denotes the Positioning of a dingo vector; and *A* is the coefficient vector. As per SADO-BM, if random integer, r > P, the update occurs as in Equation (Equation 16); otherwise, the update occurs as in Equation (Equation 17), where *BM* refers to Brownian motion. Conventionally, $\bar{b}$ is modeled as shown in Equation (Equation 18); however, as per SADO-BM, $\bar{b}$ is modeled as shown in Equation (Equation 19). Moreover, as per SADO-BM, $a_{1}$ and $a_{2}$ in Equations (20) and (21) are computed using the logistic map.
(15)$$D_{d}=\left|AP_{p}\left(x\right)-P\left(i\right)\right|$$
(16)$$P\left(i+1\right)=P_{p}\left(i\right)-B\cdot D_{d}$$
(17)$$P\left(i+1\right)=P_{p}\left(i\right)-B\cdot D_{d}+BM$$
(18)$$b=3-\left[itr\cdot *\left(\frac{3}{\max^{itr}}\right)\right]$$
(19)$$b=3\cos\left(\frac{\pi }{3}*\frac{itr}{\max^{itr}}\right)$$
(20)$$A=2\cdot a_{1}$$
(21)$$B=2\cdot b\cdot a_{2}-b$$Step 4: While *itr* < $max^{itr}$ doStep 5: The fitness is calculated using Equation (Equation 1).Step 6: The best searching agent is fixed as $d_{\alpha }$. This is presented in Equation (Equation 22).
(22)$$D_{\alpha }=\left|A_{1}\cdot P_{\alpha }---P\right|$$
(23)$$P_{1}=\left|P_{\alpha }---B\cdot D_{\beta }\right|$$Step 7: The searching agent with 2nd best searching ability is noted by $D_{\beta }$, which is presented in Equation (Equation 24).
(24)$$D_{\beta }=\left|A_{2}\cdot P_{\beta }---P\right|$$
(25)$$P_{2}=\left|P_{\beta }---B\cdot D_{\beta }\right|$$Step 8: The searching results afterwards are fixed as $D_{0}$, which is presented in Equation (Equation 26).
(26)$$D_{0}=\left|A_{3}\cdot P_{0}---P\right|$$
(27)$$P_{3}=\left|P_{0}---B\cdot D_{0}\right|$$Step 9: Iteration 1.Step 10: Repeat.Step 11: For *i* = 1:$D_{N}$ doStep 12: Renovate the position of newest searching agent.Step 13: End forStep 14: The fitness is calculated by means of Equation (Equation 1).Step 15: Compute the intensity of every dingo value of $I_{\alpha }$, $I_{\beta }$, $I_{\delta }$.Step 16: Compute $\bar{b}$, $\bar{A}$, $\bar{B}$.Step 17: *itr* = *itr* +1.Step 18: Ensure if the stopping principle has arrived.Step 19: Return $D_{\alpha }$.

## 6. Fault Tolerance and Energy Hole Mitigation

### 6.1. Fault Tolerance

The ability of a network to continue operating even when certain sensor nodes fail is known as fault tolerance. This method promises the usual operation of the network regardless of CH malfunction. Therefore, if a CH is defective, its cluster members (*cm*) become adhered to a different CH. It is divided into two groups: fault detection and fault recovery.

Fault detection: In fact *cm*, they recognize CH malfunction if they do not receive any ACK note from the CH for the transmitted data packet.

Fault recovery: During the recovery process, each cluster member *cmi* elects a novel CH with higher cost values. The proposed computation for cost is shown in Equation (Equation 28), where *yi* and *yk* denote CH and count of CH, respectively; *nccr* refers to *cm* count in present round; and *nbpr* implies count of backward CHs in preceding rounds. The term *dist* is computed in Equation (Equation 31), wherein *IEdist* and *IAdist* imply inter cluster (distance among BS and CH) and intra cluster (distance among CH and SN), respectively. In addition, $ec_{\left(yi,yk\right)}$ and $ec_{\left(yk,sink\right)}$ are computed as shown in Equations (29) and (30), in which *SNi* denotes a sensor node; $ne_{\left(SNi,yi\right)}$ and $ec_{\left(yi,sink\right)}$ imply the energy essential for transmission from the node *SNi* to *yi* and from *yi* to sink $re_{SNi}$, respectively; and $re_{yi}$ denotes the remaining energy *SNi* of *yi*.

When a sensor node is within the communication range of numerous CHs, it will therefore receive a number of messages. In this case, the sensor node chooses a CH from among its many options. We create a cost function to determine the cost value *cost*(*yi*,*yk*) of the CH *yi* for the node *yk* in order to optimize this choice. It is worth noting that a node’s selection of its specific CH from a range of available alternatives has a significant impact on the performance of WSNs. Thus, the cost function assists a node in choosing its CH among several options while considering a number of characteristics, ensuring energy efficiency and balancing.
(28)$$\mathrm{Cos}\left(yi,yk\right)=\frac{1}{eq_{{yi,yk)}^{*}}^{*}eq_{{yk,\sin k)}^{*}}^{*}ncc\left(yk\right)*nbpv\left(yk\right)*\;\mathrm{dist}\;}$$
(29)$$ec\left(SNi,yi\right)=\frac{ne_{\left(SNi,yi\right)}}{re_{SNi}}$$
(30)$$ec\left(SNi,yi\right)=\frac{ne_{\left(SNi,yi\right)}}{re_{SNi}}$$
(31)dist=IEdist+IAdist

### 6.2. Energy Hole Mitigation

An energy hole is a typical problem in sensor networks that, due to a reduced lifetime, tends to interrupt communication with the end-user. Based upon the distance of transmitting node to the sink, the sleep scheduling for every node is fixed based on $En_{req}$ that is computed by Equation (Equation 32), in which *D* denotes the packet length of data and *d* refers to the Euclidean distance among nodes; $En_{amp}$ implies power amplifier energy; and $En_{tx}$ and $En_{da}$ imply energy needed for transmission and aggregation of *D*, respectively.
(32)$$Ena_{req}=\left(\left(En_{tx}+En_{da}\right)*D\right)+\left(En_{amp}\times D*da^{*}\right)$$

## 7. Results and Discussion

### 7.1. Simulation Procedure

The proposed model for secured CHS in IoT using SADO-BM was done in MATLAB 2020a. Here, the performance of the SADO-BM scheme was proven over Cat Swarm Optimization (CSO) [19], FF Firefly (FF) [2], Shark Smell Optimization (SSO) [20], Poor Rich Optimization (PRO) [21], Hunger Games Optimizer (HGS) [22], Bald Eagle Search (BES) [23], Black Widow Optimization (BWO) [24], Fuzzy + Harris Hawks Optimization (HHO) [25], Adaptive Neuro-Fuzzy Inference System (ANFIS) + Self-Adaptive Jellyfish Search Optimizer (SA-JSO) [26], and Dingo Optimizer (DOX) [18] on wide-ranging metrics like delay, throughput, and so on. In addition, Table 3 presents the simulation parameters; this work considered 100, 250, 750, and 1000 nodes with 500, 1000, 1500, and 2000 rounds.

### 7.2. Dataset Description

The dataset for analysis was downloaded from the Hepatitis Data Set [27]. BILIRUBIN is a continuous attribute (which means that the number of its values in the ASDOHEPA.DAT file is negative); values are quoted because, when speaking about the continuous attribute, there is no such thing as all possible values. However, they represent so-called boundary values; according to these, the attribute can be discretized. At the same time, because of the continuous attribute, one can perform some other test since the continuous information is preserved.

### 7.3. Statistical Analysis

Table 4 and Table 5 reveal the statistical study for alive nodes and residual energy using the employed SADO-BM model over conventional models (CSO, FF, SSO, PRS, HGS, BES, BWO, Fuzzy + HHO, ANFIS + SA-JSO, and DOX). “The met heuristic schemes are stochastic, and to substantiate its fair evaluation, each model is analyzed quite a lot of times to accomplish Equation (Equation 1)”. In Table 4, the proposed SADO-BM method has attained higher alive node counts (900.26) for the mean case for 1000 nodes. Among the schemes, conservative SSO achieved the worst values for every scenario compared to CSO, FF, PRS, HGS, BES, BWO, Fuzzy + HHO, ANFIS + SA-JSO, and DOX. The proposed work achieves enhanced fault tolerance and energy hole mitigation. This is owing to the deployment of hybrid optimization named SADO-BM, which optimally chooses the CH in the IoT network.

### 7.4. Convergence Analysis

The convergence of the SADO-BM method over CSO, FF, SSO, PRS, HGS, BES, BWO, and DOX) for diverse iterations is depicted in Figure 3. Essentially, the SADO-BM has accomplished the lowest cost values with the increase in the iteration for all nodes. As shown in Figure 3b, from the 8th to the 10th iteration, the values for cost have reduced to 4 for 250 nodes. Likewise, in Figure 3c, at the 10th iteration, the cost has condensed to 2 for 750 nodes. The other evaluated schemes like CSO, FF, SSO, PRS, HGS, BES, BWO, and DOX revealed relatively high cost values. Therefore, with the SADO-BM–based optimization, better results are attained for clustering.

### 7.5. Analysis of Delay and Distance

The analysis on distance and delay using the SADO-BM method over CSO, FF, SSO, PRS, HGS, BES, BWO, Fuzzy + HHO, ANFIS + SA-JSO, and DOX is interpreted in Figure 4 and Figure 5. The assessment was done for varied nodes (100, 250, 750, and 1000). In fact, the distance between the CH and BS and the delay to transfer packets have to be the lowest in order to achieve superior performance. If the distances increase, then automatically, the delay also increases. Here, the delay is done in seconds. In Figure 4, the distance values fluctuate for varied rounds from 0 to 2000. At first, for 100th, the distance using SADO-BM at the 1000th round is high (10.5 × $10^{4}$). For other rounds, the values of distance are low using SADO-BM. Similarly, the delay values fluctuate for varied rounds from 0 to 2000. Nevertheless, the SADO-BM method is superior over the CSO, FF, SSO, PRS, HGS, BES, BWO, Fuzzy + HHO, ANFIS + SA-JSO, and DOX models. These enhancements are due to the enhanced fault tolerance and energy hole mitigation concepts.

### 7.6. Analysis on Alive Nodes

The study of the SADO-BM approach relating to the alive node is illustrated in Figure 6. The development of SADO-BM was established over CSO, FF, SSO, PRS, HGS, BES, BWO, Fuzzy + HHO, ANFIS + SA-JSO, and DOX. Here, assessment is done for 100, 250, 750, and 1000 nodes. The count of alive nodes must be high for higher performance. In Figure 6, the counts of alive nodes are lowered with the increase in rounds. In Figure 6b, at the 0–1000th round, the alive nodes for SADO-BM are 250, while, from the 1000th to the 2000th round, the alive nodes start lessening for SADO-BM and reach 160. Similarly, in Figure 6d, at the 0–1000th round, the alive nodes for SADO-BM are 1000, whereas at the 1000–2000th round, the alive nodes start lessening for SADO-BM and reach 700. However, the SADO-BM shows higher alive nodes than CSO, FF, SSO, PRS, HGS, BES, BWO, Fuzzy + HHO, ANFIS + SA-JSO, and DOX.

### 7.7. Analysis on Energy and Overhead

The energy and overhead using the SADO-BM technique over CSO, FF, SSO, PRS, HGS, BES, BWO, Fuzzy + HHO, ANFIS + SA-JSO, and DOX are represented here. The higher the leftover energy, the better the system performance. This shows that the selection of CH is done with minimum energy. In Figure 7a, the remaining energy reduces with an increase in the rounds. The remaining energy for SADO-BM at the 0th round is 0.55, while for the 2000th round, the remaining energy is 0.275 for the 1000th node. Similarly, in Figure 8, the overhead is lower at the 2000th round for all counts of nodes (100, 250, 750, and 1000). The other compared models (CSO, FF, SSO, PRS, HGS, BES, BWO, Fuzzy + HHO, ANFIS + SA-JSO, and DOX) present higher overhead, while the SADO-BM model shows less overhead.These enhancements are due to the enhanced fault tolerance and energy hole mitigation in the adopted SADO-BM theory.

### 7.8. Analysis on Throughput and Trust

The analyses on throughput and trust using the SADO-BM method over CSO, FF, SSO, PRS, HGS, BES, BWO, Fuzzy + HHO, ANFIS + SA-JSO, and DOX are established in Figure 9 and Figure 10, respectively. Here, the assessment is made for 100, 250, 750, and 1000 nodes. “Throughput is a measure of how many units of information a system can process in a given amount of time. Throughput is usually measured in bits per second (bit/s or bps), and sometimes in data packets per second (p/s or pps) or data packets per time slot”. The throughput must be higher as it plays a lead role in data transferring. In Figure 9a, the throughput using SADO-BM at primary stages is high, while, as rounds increase, the throughput values are dropped slightly. Likewise, in the case of trust, the results vary for all rounds. However, the SADO-BM has attained higher trust over CSO, FF, SSO, PRS, HGS, BES, BWO, Fuzzy + HHO, ANFIS + SA-JSO, and DOX. This perceptibly promised better performance of SADO-BM with enhanced fault tolerance and energy hole mitigation.

## 8. Conclusions

This work introduced a cluster-based approach in the Internet of Things (IoT). This work deployed Self-Adaptive Dingo Optimizer with Brownian Motion (SADO-BM) for clustering by considering “energy, distance, delay, overhead, trust, Quality of service (QoS), and security (high risk, low risk, and medium risk)”. Then, if the optimal Cluster Head (CH) contained any defects, fault tolerance as well as energy hole mitigation were performed. From the outcomes, the distance values fluctuated for varied rounds from 0 to 2000. Firstly, for the 100th node, the distance using SADO-BM at the 1000th round was high (10.5 × $10^{4}$). For other rounds, the values of distance were low using SADO-BM. Similarly, the delay values fluctuated for varied rounds from 0 to 2000. Nevertheless, the SADO-BM method was superior over Cat Swarm Optimization (CSO), Firefly (FF), Shark Smell Optimization (SSO), Poor Rich Optimization (PRO), Hunger Games Optimizer (HGS), Bald Eagle Search (BES), Black Widow Optimization (BWO), Fuzzy + Harris Hawks Optimization (HHO), Adaptive Neuro-Fuzzy Inference System (ANFIS) + Self-Adaptive Jellyfish Search Optimizer (SA-JSO), and Dingo Optimizer (DOX). The cluster-based routing in IoT has certain benefits over wired systems, including ease of use, lower delivery costs, reduced risk of failures, and increased mobility. In the future, studies of this paper can be extended to work in Social IoT and Multiple IoT scenarios.

## Figures and Tables

**Figure 1 sensors-22-08064-f001:**
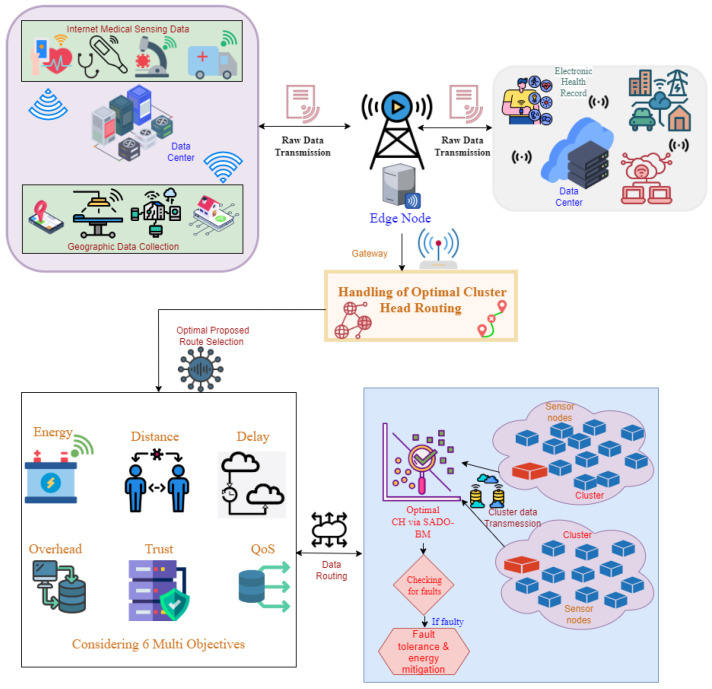
Representation of the proposed model.

**Figure 2 sensors-22-08064-f002:**
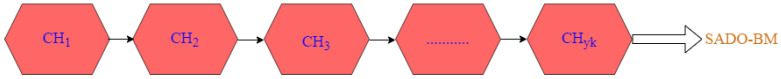
Solution encoding.

**Figure 3 sensors-22-08064-f003:**
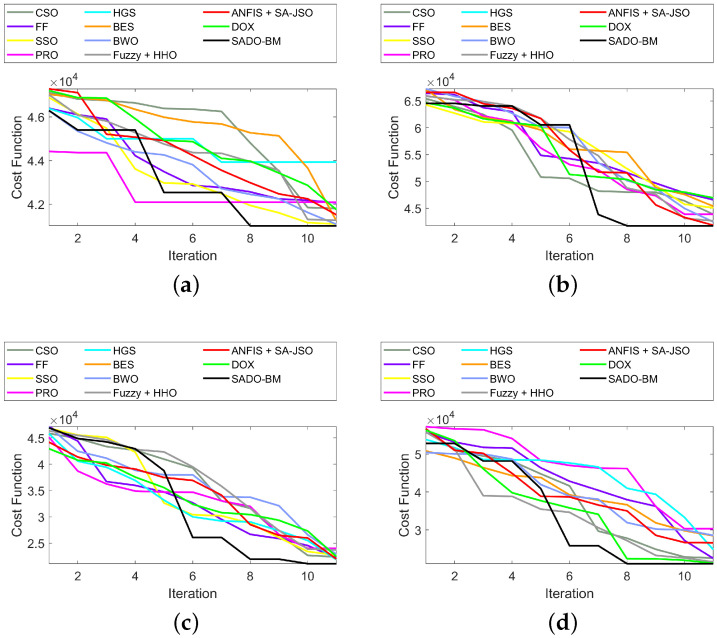
Convergence of SADO-BM over other models for node counts (**a**) 100, (**b**) 250 (**c**) 750, and (**d**) 1000.

**Figure 4 sensors-22-08064-f004:**
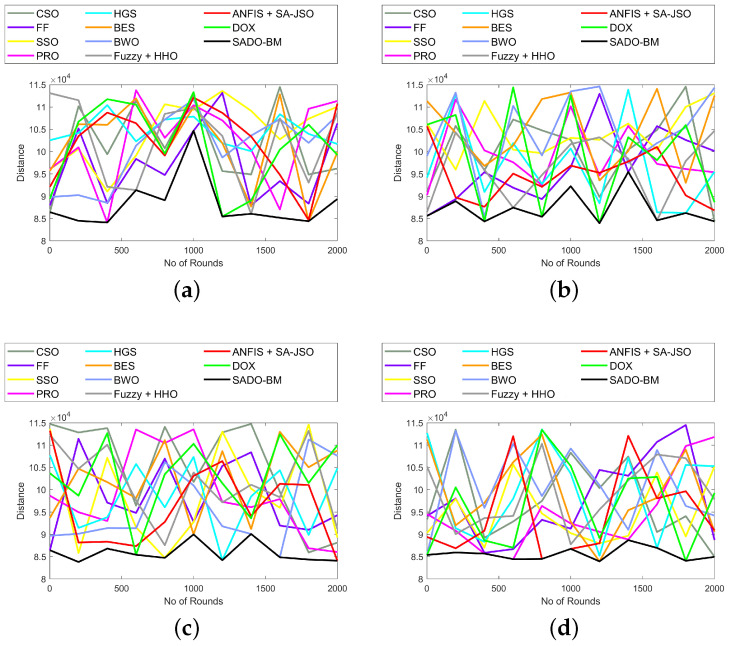
Distance analysis of SADO-BM over other models for node counts (**a**) 100, (**b**) 250 (**c**) 750, and (**d**) 1000.

**Figure 5 sensors-22-08064-f005:**
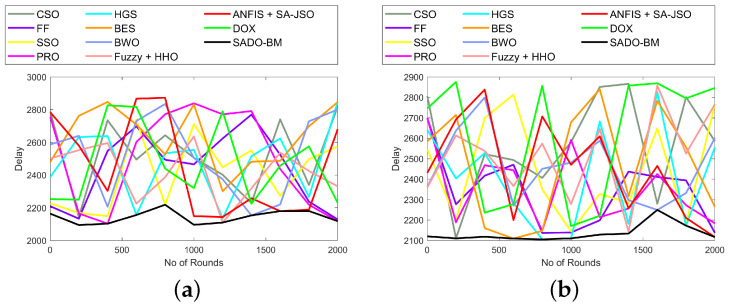
Delay analysis of SADO-BM over other models for node counts (**a**) 100, (**b**) 250 (**c**) 750, and (**d**) 1000.

**Figure 6 sensors-22-08064-f006:**
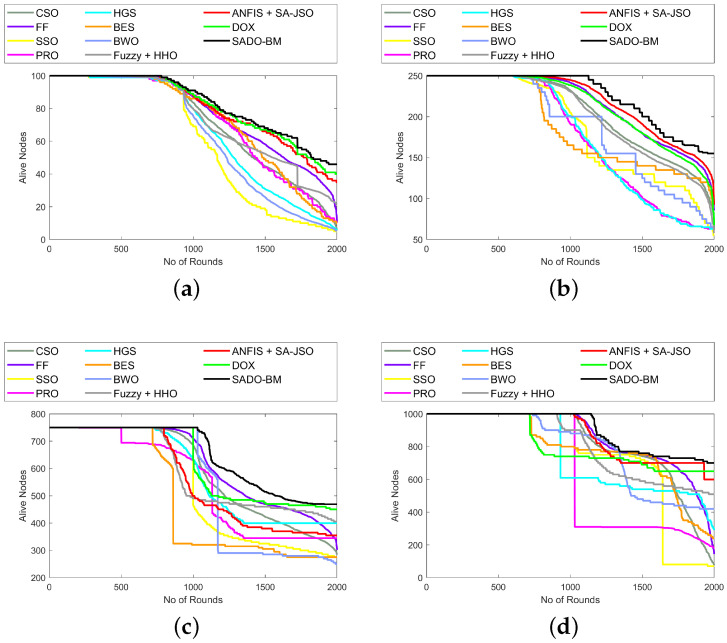
Alive node analysis of SADO-BM over other models for node counts (**a**) 100, (**b**) 250 (**c**) 750, and (**d**) 1000.

**Figure 7 sensors-22-08064-f007:**
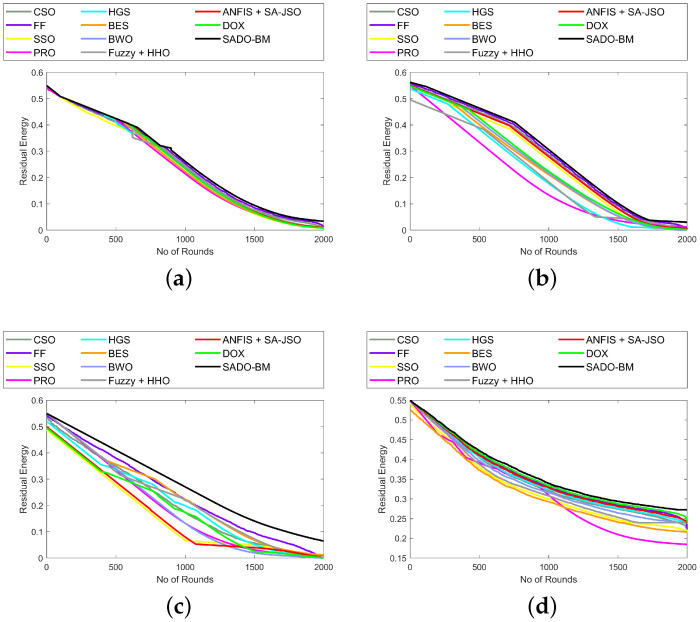
Energy analysis of SADO-BM over other models for node counts (**a**) 100, (**b**) 250 (**c**) 750, and (**d**) 1000.

**Figure 8 sensors-22-08064-f008:**
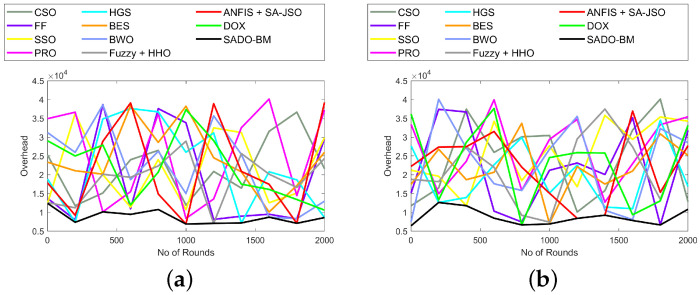
Overhead analysis of SADO-BM over other models for node counts (**a**) 100, (**b**) 250 (**c**) 750, and (**d**) 1000.

**Figure 9 sensors-22-08064-f009:**
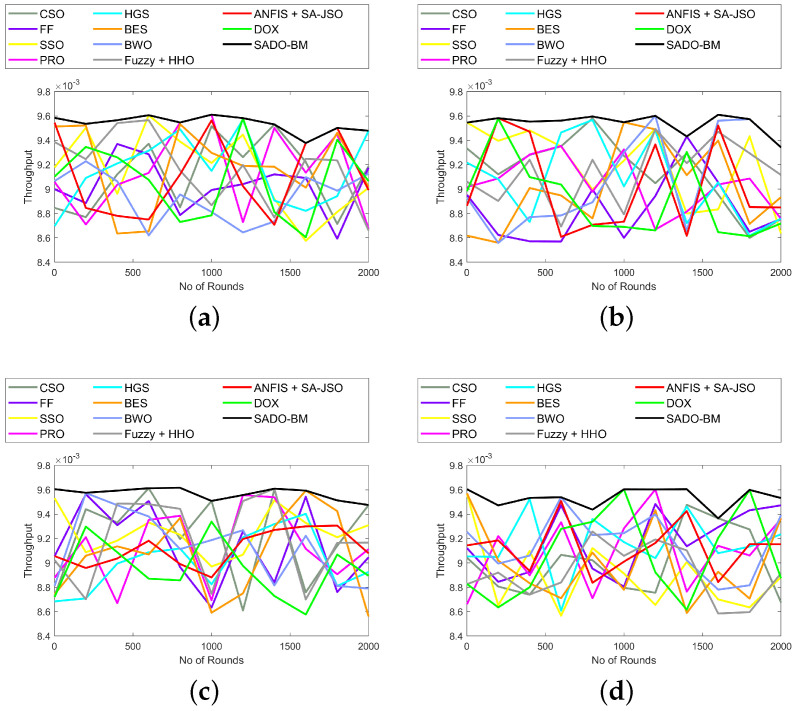
Throughput analysis of SADO-BM over other models for node counts (**a**) 100, (**b**) 250 (**c**) 750, and (**d**) 1000.

**Figure 10 sensors-22-08064-f010:**
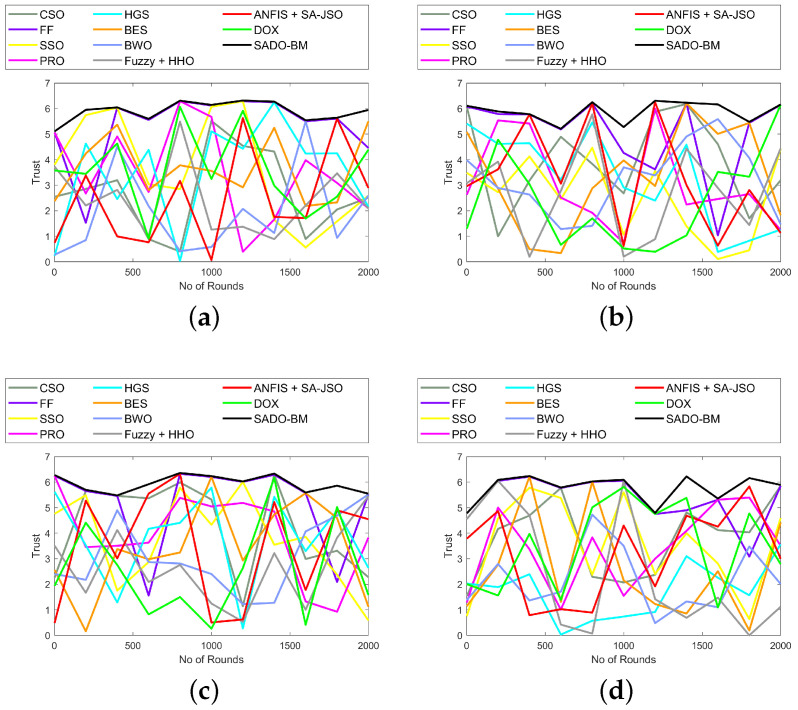
Trust analysis of SADO-BM over other models for node counts (**a**) 100, (**b**) 250 (**c**) 750, and (**d**) 1000.

**Table 1 sensors-22-08064-t001:** Nomenclature.

Abbreviation	Description
ANFIS	Adaptive Neuro-Fuzzy Inference System
DoS	Denial Of Service
EMGR	Energy-Efficient Multicast Geographic Routing Protocol
EER	Energy-Efficient Routing
EEG	Energy-Efficient Geographic
FGSA	Fractional Gravitational Search Algorithm
GPS	Global Positioning System
GWO	Grey Wolf Optimizer
IoT	Internet Of Things
MOFGSA	Multi-Objective FGSA
MCCs	Multi-Hop Communication Cells
MSE	Mean Square Error
PDR	Packet Delivery Ratio
SFG	Sunflower Based GWO
SNs	Sensor Nodes
SelGOR	Selective Authentication-Based Geographic Opportunistic Routing
SSI	Statistic State Information
SCCs	Single-Hop Communication Cells
SA-JSO	Self-Adaptive Jellyfish Search Optimizer
SFO	Sun Flower Optimization
QoS	Quality Of Service
WSN	Wireless Sensor Network

**Table 2 sensors-22-08064-t002:** Review of conventional IoT protocol.

Ref. No	Proposed Model	Pros	Cons
1	HiLSeR	High throughput Enhances PDR	Low PDR High end-to-end delay
2	Bagging Classifier	High accuracy Minimal execution time	Cost-optimal execution is not considered.
3	ANN	High specificity Enhanced accuracy	Compressive sensing should be more concerned.
4	EGRPM	Maximizes life span Minimizes delay	No consideration on overhead.
5	EEL routing	Low error High PDR	More consumption of energy.
6	ELBAR	Improved lifespan of the network Average energy utilization	Need spotlight on numerous holes.
7	Fuzzy	Less distance amid hops Low energy utilization	Require evaluation on communication costs.
8	IoGHR	More residual energy Higher life span	Several mobile sinks should be deployed.

**Table 3 sensors-22-08064-t003:** Simulation parameters.

Channel Type	Wireless
Antenna	Omni Antenna
Dimension X	100 m
Dimension Y	100 m
Total Simulation time	10 s
Number of nodes	100, 250, 750, 1000

**Table 4 sensors-22-08064-t004:** Statistical Analysis for SADO-BM Over Other Models Regarding Alive Nodes.

Node = 100											
Measures	CSO	FFO	SSO	PRS	HGS	BES	BWO	Fuzzy + HHO	ANFIS + JSO	DOX	SADO-BM
Min	6.0122	10.668	5	10	5.5793	10	5.076	21.409	35	40	46
Max	100	100	100	100	100	100	100	100	100	100	100
Mean	72.208	77.797	60.88	73.045	66.902	73.57	64.445	74.08	80.829	81.88	83.259
Median	83.459	86.977	69	88	79.722	86	75.462	79.689	88	89	91
STD	29.648	24.057	38.608	29.857	34.181	29.928	36.278	26.546	20.834	19.586	18.324
Node = 250											
Min	61.896	85.527	55	63	63	90	65	57.904	92.638	67.541	155
Max	250	250	250	250	250	250	250	250	250	250	250
Mean	206.28	215.85	186.31	177.09	177.24	187.81	187.93	203.12	219.95	214.17	225.83
Median	231.93	241.37	205	191	201	165	200	230.62	244.78	238.28	250
STD	48.437	41.128	62.495	72.97	73.731	52.679	62.114	51.181	38.22	42.85	33.354
Min	61.896	85.527	55	63	63	90	65	57.904	92.638	67.541	155
Node = 750											
Min	284.54	301.61	275	345	400	275	250	405	355	450	469
Max	750	750	750	750	750	750	750	750	750	750	750
Mean	590.81	618.79	535.25	550.83	584.25	487.08	545.82	586.84	560.73	616.93	647.74
Median	684.4	709.91	465	630	630	320	750	490	500	630	750
STD	166.22	142.36	205.29	175.15	159.54	215.3	224.87	140.78	170.83	134.81	117.63
Node = 1000											
Min	75.92	144	70	182	290	240	420	510	600	650	700
Max	1000	1000	1000	1000	1000	1000	1000	1000	1000	1000	1000
Mean	824.38	862.05	753.79	655.82	756.33	790.75	788.36	810.46	873.16	812.82	900.26
Median	1000	1000	1000	1000	610	800	885	900	1000	740	1000
STD	242.38	184.91	336.87	355.54	231.78	218.93	233.25	199.29	145.2	145.7	121.63

**Table 5 sensors-22-08064-t005:** Statistical Analysis for SADO-BM over Other Models Regarding Residual Energy.

Node = 100											
Measures	CSO	FFO	SSO	PRS	HGS	BES	BWO	Fuzzy + HHO	ANFIS + JSO	DOX	SADO-BM
Min	0.008224	0.012222	0.006799	0.010732	0.007795	0.010444	0.010196	0.011195	0.010523	0.008526	0.034196
Max	0.54958	0.54958	0.54958	0.54958	0.54916	0.54958	0.54958	0.54957	0.54958	0.54958	0.54958
Mean	0.2502	0.25688	0.23647	0.2391	0.24547	0.24253	0.24846	0.24996	0.24744	0.24658	0.26377
Median	0.24285	0.25384	0.22276	0.21443	0.23522	0.22274	0.23755	0.24535	0.23405	0.23347	0.26155
STD	0.1708	0.16922	0.17355	0.17662	0.17594	0.17811	0.17703	0.17445	0.17753	0.17819	0.16741
Node = 250											
Min	0.000106	0.003215	2.31 × $10^{-5}$	0.012439	1.71 × $10^{-5}$	1.82 × $10^{-6}$	6.23 × $10^{-6}$	2.22 × $10^{-5}$	0.007754	0.002181	0.030434
Max	0.55895	0.56138	0.53727	0.54957	0.53715	0.54958	0.54958	0.49525	0.54958	0.54727	0.56188
Mean	0.27383	0.28288	0.25741	0.19202	0.21328	0.23381	0.23826	0.21268	0.26926	0.24415	0.29136
Median	0.28424	0.29567	0.26709	0.13372	0.17706	0.21101	0.21647	0.18311	0.2794	0.22338	0.30624
STD	0.18503	0.18469	0.18378	0.16914	0.1841	0.18385	0.18589	0.16916	0.18443	0.18431	0.18423
Node = 750											
Min	7.27 × $10^{-5}$	5.45 × $10^{-5}$	0.000106	0.010441	0.00044	0.012122	8.75 × $10^{-6}$	0.00267	1.84 × $10^{-5}$	4.08 × $10^{-5}$	0.064935
Max	0.5419	0.54774	0.48698	0.54957	0.51958	0.54958	0.54959	0.54957	0.49929	0.49447	0.54972
Mean	0.2079	0.24185	0.15963	0.19055	0.20786	0.22942	0.18809	0.22166	0.16268	0.18766	0.28055
Median	0.173	0.223	0.06766	0.1322	0.19929	0.22268	0.13227	0.21938	0.07997	0.17418	0.27058
STD	0.16152	0.15846	0.14612	0.17098	0.15697	0.16253	0.17634	0.15992	0.15356	0.15107	0.14798
Node = 1000											
Min	0.22118	0.22482	0.21714	0.18466	0.23974	0.21625	0.23965	0.23965	0.23996	0.24123	0.27253
Max	0.54914	0.54923	0.54913	0.54957	0.54958	0.52618	0.54958	0.54958	0.54958	0.54958	0.54958
Mean	0.34496	0.35491	0.32444	0.3152	0.34864	0.31685	0.34025	0.33218	0.35267	0.35885	0.36471
Median	0.3205	0.32933	0.29959	0.30941	0.32386	0.29246	0.31586	0.30774	0.32863	0.33358	0.33981
STD	0.086216	0.083868	0.089364	0.10569	0.086094	0.088345	0.088345	0.090045	0.084979	0.083003	0.081136

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
