# Peer review of "Dingo Optimization Based Cluster Based Routing in Internet of Things"

_sensors, 2022, doi:10.3390/s22208064_

Round 1

Reviewer 1 Report

This paper is hard to read.

Namely, paper novelties, benefits are not noted. Introduction need to gove answers on these points. For that reason please rewrite paper and write it on standard manner. Also, add paper organisation. Section with mathematic equation is very poor described. Also, note all used variables and describe used map. Section with results os written in hard manner for tracking. Conclusion section is writtem in hard manner with a lot of short words (abbrevations) and therefore it is hard to understand concept. Literature list must contain a referebcies from mdpi journals. 

Author Response

This paper is hard to read.

Response:  Thanks very much for your comment.  As per the abovementioned query, proof read has been made in the revised manuscript.

Namely, paper novelties, benefits are not noted. Introduction need to gove answers on these points.

Response:  Thanks very much for your comment. As per the abovementioned query, the paper novelties have been clearly presented in 7th line of the abstract and in the 4th paragraph of the introduction section.

Update in Abstract:

As a novelty, this work intends to introduce a cluster-based approach in IoT. For this a Self-Adaptive Dingo Optimizer with Brownian Motion (SADO-BM) for clustering is introduced for optimally selecting the CH by considering energy, distance, delay, overhead, trust, Quality of Service (QoS), and security (high risk, low risk and medium risk). If optimal CH contains any defects, fault tolerance, as well as energy hole mitigation, is performed. Eventually, analysis is done to ensure the progression of the SADO-BM model.

Updates in Section I:

The main contributions are as follows:

  • This work deploys cluster-based routing in the internet of things.
  • To optimally elect the CH in the IoT network, a new hybrid optimization named SADO-BM is introduced.
  • Exploits fault tolerance and energy hole mitigation process if the optimal CH is found to be a defective one.

For that reason please rewrite paper and write it on standard manner. Also, add paper organisation.

Response:  Thanks very much for your comment. As per your suggestion paper has been rewritten in the standard manner. Moreover, the organization has been added in the 5th paragraph of the revised manuscript.

Updates in Section I:

The paper's organization is as follows: The review on a relevant topic was analyzed in Section 2. Section 3 narrates the general idea of IoT in healthcare appliances. Section 4 and Section 5 depict about objectives and development of SADO-BM. Section 6 describes fault tolerance and energy hole mitigation. The results and conclusions are given in Sections 7 and 8.

Section with mathematic equation is very poor described. Also, note all used variables and describe used map.

Response:  Thanks very much for your comment. As per the abovementioned query, all used variables are defined.

Section with results os written in hard manner for tracking.

Response:  Thanks very much for your comment. As per your suggestion, the results and discussion section have been revised.

Conclusion section is writtem in hard manner with a lot of short words (abbrevations) and therefore it is hard to understand concept.

Response:  Thanks very much for your comment. All the abbreviations are defined in the conclusion section.

Updates in Conclusion:

Cat Swarm Optimization (CSO), Firefly (FF), Shark Smell Optimization (SSO), Poor Rich Optimization (PRO), Hunger Games Optimizer (HGS), Bald Eagle Search (BES), Black Widow Optimization (BWO), Fuzzy + Harris Hawks Optimization (HHO), Adaptive Neuro-Fuzzy Inference System (ANFIS)+ Self-Adaptive Jellyfish Search Optimizer (SA-JSO) and Dingo Optimizer (DOX).

Reviewer 2 Report

In this paper, the authors introduce a cluster-based approach in IoT. Specifically, the proposed approach deploys SADO-DM for clustering and, if necessary, performs fault tolerance and hole mitigation.

The topic considered by the authors is interesting. The approach appears to be well set up from a technical point of view. In addition, the authors illustrate many experiments they have conducted to validate their approach.

My main concern about this paper is the general presentation of the approach. That presentation also appears confusing because the English style used by the authors is rich of errors. First of all, I suggest the authors to get the paper proofread by a native English speaker.

As a second suggestion, I think the single and classical IoT context considered by the authors in the paper is a bit outdated. The authors should add a session "Discussion" after the experiments to indicate how their approach can be extended to more innovative IoT architectures proposed in the literature, such as the SIoT (Social IoT), proposed by Iera, Morabito et al. and the MIoT (Multiple IoT), proposed by Cauteruccio, Virgili et al.

Author Response

In this paper, the authors introduce a cluster-based approach in IoT. Specifically, the proposed approach deploys SADO-DM for clustering and, if necessary, performs fault tolerance and hole mitigation.

The topic considered by the authors is interesting. The approach appears to be well set up from a technical point of view. In addition, the authors illustrate many experiments they have conducted to validate their approach.

Response:  Thanks very much for your comment. Thank you for your positive response.

My main concern about this paper is the general presentation of the approach. That presentation also appears confusing because the English style used by the authors is rich of errors. First of all, I suggest the authors to get the paper proofread by a native English speaker.

Response:  Thanks very much for your comment. As per the abovementioned query, proofread has been made in the revised manuscript.

As a second suggestion, I think the single and classical IoT context considered by the authors in the paper is a bit outdated. The authors should add a session "Discussion" after the experiments to indicate how their approach can be extended to more innovative IoT architectures proposed in the literature, such as the SIoT (Social IoT), proposed by Iera, Morabito et al. and the MIoT (Multiple IoT), proposed by Cauteruccio, Virgili et al.

Response:  Thanks very much for your comment. As you instructed, we have added discussion about the above mentioned content at the end of the conclusion section. Moreover the Iera, Morabito et al and Cauteruccio, Virgili et al have been cited in the introduction section.

Updates in Conclusion:

In the future, studies this paper can be extended to work in Social IoT and Multiple IoT scenarios.

Round 2

Reviewer 1 Report

This paper is still hard to read. Furthermore, the Abstract is unclear  - added text needs to be checked, and improved!

The introduction is not improved - the authors did not answer my questions. This is not a scientific style for abstract! Please see some papers on how to write Introduction.

Author Response

This paper is still hard to read. Furthermore, the Abstract is unclear  - added text needs to be checked, and improved!

Response: Thank you for your comment we have modified the abstract. 

The introduction is not improved - the authors did not answer my questions. This is not a scientific style for abstract! Please see some papers on how to write Introduction.

Response: Thank you for your comment we have modified the introduction.  

Reviewer 2 Report

The authors have complied with all my previous requests. As a consequence, I think that the paper can be accepted in its current version.

Author Response

The authors have complied with all my previous requests. As a consequence, I think that the paper can be accepted in its current version.

Response: Thank you for your positive feedback. 

Round 3

Reviewer 1 Report

No more comments.